# Temporally programmed polymer – solvent interactions using a chemical reaction network

Benjamin Klemm [1], Reece W. Lewis [1], Irene Piergentili [1] & Rienk Eelkema [1] ✉

Out of equilibrium operation of chemical reaction networks (CRNs) enables artificial materials to autonomously respond to their environment by activation and deactivation of intermolecular interactions. Generally, their activation can be driven by various chemical conversions, yet their deactivation to non-interacting building blocks remains largely limited to hydrolysis and internal pH change. To achieve control over deactivation, we present a new, modular CRN that enables reversible formation of positive charges on a tertiary amine substrate, which are removed using nucleophilic signals that control the deactivation kinetics. The modular nature of the CRN enables incorporation in diverse polymer materials, leading to a temporally programmed transition from collapsed and hydrophobic to solvated, hydrophilic polymer chains by controlling polymer-solvent interactions. Depending on the layout of the CRN, we can create stimuli-responsive or autonomously responding materials. This concept will not only offer new opportunities in molecular cargo delivery but also pave the way for next-generation interactive materials.

Interactive materials, which are able to adapt and to interact with their surroundings by responding to events taking place in their environment[1], will find many applications ranging from regenerative medicine, optoelectronics to nanomachines[2–4]. Response to environmental cues through controlled material growth and decay is a unique property, which enables nature to perform complex functions, for example signal transduction, cell-division and intracellular transport[5,6]. Here, the conversion of chemical fuels such as adenosine triphosphate (ATP) are used to temporally control out-of-equilibrium assembly into supramolecular structures[7].

Synthetic analogues of these active materials can be designed by coupling the activation of non-interacting building blocks to fuel-driven chemical reaction networks (CRNs)[8]. Crucially, the availability of chemical fuel sustains the active material, while depletion of the fuel causes the material to disassemble back to its precursor[9]. Besides strategies that involve activation of the surroundings, such as a transient pH change[10–13], a widely applied strategy is the direct activation of building blocks by a chemical fuel[9]. In such systems, the chemical fuel reacts with an inactive building block, converting it to an activated intermediate (activation). A second reaction subsequently converts the intermediate to its inactive precursor state by spontaneously forming a waste product (deactivation)[14].

A wide variety of material activation processes have been described in literature, which are typically powered by various chemical fuels (e.g. carbodiimide fuels[15], methylation agents[5], redox-reagents[16,17] or thioesters[18]) or physical stimuli, such as light or ultrasound to name a few[19–21]. In contrast, the deactivation mechanism (excluding enzymatic reactions) for chemically fuelled non-equilibrium CRNs at present relies frequently on hydrolysis[9,15,22–25] or internal pH change[10–13]. Notable exceptions[26,27] concern deactivation reactions that use either redox chemistry[17,28–30], alkene cross-metathesis[31], chemical clocks[32] or spontaneous chemical degradation[18].

This work describes the development and application of a new type of CRN, based on nucleophilic substitution by using allylic electrophiles as chemical fuel. This class of CRNs is characterized by a wide range of reactivities and substrate structures[33,34], rendering it highly

[1]Department of Chemical Engineering, Delft University of Technology, Van der Maasweg 9, 2629 HZ Delft, The Netherlands. ✉e-mail: R.Eelkema@tudelft.nl

versatile and potentially widely applicable. In particular, the wide range of nucleophilic reactivities and their prevalence in nature opens the door to precise nucleophilic control over the deactivation step, enabling highly tuneable and responsive CRNs.

So far, studies of transient materials have focused on applying existing CRNs to different material classes, frequently composed of low molecular weight amphiphiles[20,32,35,36], gelator assemblies[10,11,29], nanoparticles[22] or (block)-copolymers[13,16,17,23,24,37–41]. In particular for polymer materials, established (de)activation strategies are generally used (e.g., pH change, carbodiimide/hydrolysis, redox-reagents or light). In contrast, new CRNs are commonly developed for a specific material class or connected to the design of the molecular building blocks. In consideration of current methods for the development of materials with autonomous behaviour and pre-programmed response, we lack a general framework[42]. Thus, the development of scalable CRNs and their applicability to a variety of different materials for the design of interactive structures is an extremely attractive challenge.

In this work, we introduce a general strategy towards developing interactive synthetic materials by using a CRN that can be applied to polymers of varying structure and composition, and is scalable from small molecule models (molecular level) to nanoscale supramolecular materials (block copolymer micelles) to macroscale macromolecular crosslinked superstructures (polymer hydrogels). These materials undergo a temporally programmed transition from collapsed, hydrophobic polymer chains to solvated, hydrophilic polymer domains by controlling polymer-solvent interactions. We can achieve stimuli-responsive or autonomous material changes by controlling these interactions using our CRN strategy to reversibly form positive charges along a polymer backbone in aqueous buffer, at physiological pH.

## Results and discussion

The CRN central to this work is based on the allylic substitution of electron deficient Morita-Baylis-Hillman allyl acetates with tertiary nitrogen nucleophiles. In this reaction, a metastable, positively charged quaternary nitrogen adduct is formed[43,44]. While previous studies have rationalized the formation of this adduct[45,46] or isolated their bromine and chlorine counterion – salts from non-aqueous solutions, we discovered that the formation of the acetate counterion – adduct (activated intermediate) is stable in buffered aqueous solution at neutral pH and room temperature. Subsequent addition of competing S or N-terminal nucleophiles initiates a second allylic substitution on the activated intermediate, which reverses the quaternary nitrogen to the neutral amine, forming the allylic reaction product (waste) and thus completing the reaction cycle (Fig. 1a). Both the substitution of the tertiary amine on the allyl acetate as well as the subsequent substitution of the nucleophile on the quaternary amine intermediate likely proceed via conjugate additions followed by elimination of the allylic leaving group from the enolate, mechanisms that are related to E1cB type eliminations and the Morita-Baylis-Hillman reaction.

In the newly developed CRN, the allyl substrate diethyl(α-acetoxymethyl) vinylphosphonate (DVP)[47] acts as a fuel, enabling a reversible switch between charge states of the nitrogen centre. We are able to manipulate the CRN and the subsequent material response by delicate design of the allylation reaction (activation) and its successive substitution reaction in the presence of nucleophiles (deactivation). Highly nucleophilic thiols in water[48,49], such as 2-mercaptoethanol (SH-**3**, Fig. 1b), react first with the activated intermediate before attacking the fuel itself. This allows for signal-controlled cycling between 'charged' and 'uncharged' species, referred to as signal-induced cycling (Fig. 1a-right). We studied two common tertiary amines (Fig. 1b) in a small molecule CRN: DABCO (t-Am-**1**) and pyridine (t-Am-**2**) and evaluated their behaviour in the CRN. Furthermore, we investigated the potential of chemically fuelled, out-of-equilibrium systems, where competition of (de)

activation reactions leads to autonomous cycling (Fig. 1a-right). Here, we sought for weak nucleophiles, which were not reactive with the fuel. Amino acids such as threonine (NH$_2$-**4**, Fig. 1b) are prime candidates, due to their weak nucleophilicity (nucleophilicity index ($N$) = 12.69 in water[50]). The primary amine pK$_a$ of threonine is 9.1 meaning that at neutral pH, it is mostly present as the protonated amine species (> 97%)[51]. Under optimised conditions, this allows for an initial accumulation of the charged activated intermediate, which is later reversed by the delayed substitution from excess NH$_2$-**4** to form the uncharged waste products (DVP-N + DVP-2-N, Fig. 1b).

To bridge this concept towards synthetic materials, we synthesised block and statistical copolymers of N,N-dimethylacrylamide and 4-vinylpyridine (P1 and P2, Fig. 1c-left) by a two-step reversible addition fragmentation chain-transfer (RAFT) process[52]. The pK$_a$ of poly(4-vinylpyridine) is 5.0 ± 0.3[53] meaning that at neutral pH the free base species is dominant, making it an excellent candidate for reversible charge formation in neutral pH buffered aqueous systems. With those polymers in hand, we prepared two types of materials: (1) supramolecular micellar dispersions and (2) macromolecular polymer hydrogels (Fig. 1c-right). Fuelling these materials with DVP, we anticipated polymer dissolution in micellar dispersions by induced hydrophilization and polymeric network expansion within hydrogels by osmotic pressure forced water intake. The subsequent decay of the intermediate species will convert the positive charge along the polymer, and in turn lead to micellar re-formation and hydrogel contraction due to the loss of charges and osmotic pressure[54] (Fig. 1d).

### Signal-induced and autonomous small-molecule model CRN

In the small molecule CRN (Fig. 2a), amine substrates t-Am-**1** or t-Am-**2** react with DVP to generate an intermediary allylammonium ion (DVP-t-Am-**1** or DVP-t-Am-**2**). After fuel activation, reaction with a thiol (SH-**3**) forms a waste product (DVP-S) and regenerates the amine substrate, thus completing one cycle. Using sub-stoichiometric amounts of amine (0.2 eq.) with excess fuel (1.0 eq.) we first converted t-Am-**1** and t-Am-**2** to the activated intermediates (0.2 eq.), which upon SH-**3** addition (0.2 eq.) reversed back to the neutral amine. The remaining unreacted fuel (~0.8 eq.) then spontaneously regenerates the activated intermediate, which allows for continuous cycling upon consecutive thiol additions. Using $^1$H NMR spectroscopy, we confirmed four consecutive reaction cycles by controlled deionization of the activated intermediate with 4x SH-**3** additions (0.2 eq.) at stable pH conditions (Fig. 2b, c). We conducted a reaction rate study to further understand the reactivity of DVP towards the tertiary amines and nucleophiles with their order being t-Am-**1** > SH-**3** > t-Am-**2** » NH$_2$-**4** (Supplementary Fig. 2). As t-Am-**1** ($N$ = 18.80 in CH$_3$CN[55]) is more nucleophilic than t-Am-**2** ($N$ = 12.90 in CH$_2$Cl$_2$[55]), we can attribute these kinetic variations to the difference in nucleophilicity of the employed tertiary amine[56]. Hence, it was not surprising that complete conversion of DVP to DVP-t-Am-**1** is on average ~9.0 ± 0.8 fold faster than that of DVP-t-Am-**2** (Fig. 2b, c). Similarly, for the progression of the deactivation reaction, DVP-S formed on average ~30 ± 3.4 times faster using t-Am-**1** than when using t-Am-**2** (Fig. 2b, c). The blank reaction of DVP with thiol (no tertiary amine) using a ratio of 1:1, takes ~110 h to reach completion (Supplementary Figs. 3 & 5). Addition of thiol to a mixture of activated intermediate in excess DVP predominantly leads to reaction with the activated intermediate (Fig. 2b, c). This observation confirms that the reactivity of thiol with the activated intermediate (DVP-t-Am-**1** or DVP-t-Am-**2**) is kinetically highly favoured over the background reaction with DVP. Although reaction kinetics are amine dependent, in both cases the formation of waste product follows quantitatively after each SH-**3** addition event (Fig. 2b, c), which confirms the absence of unwanted SH-**3** side reactivity such as disulfide formation.

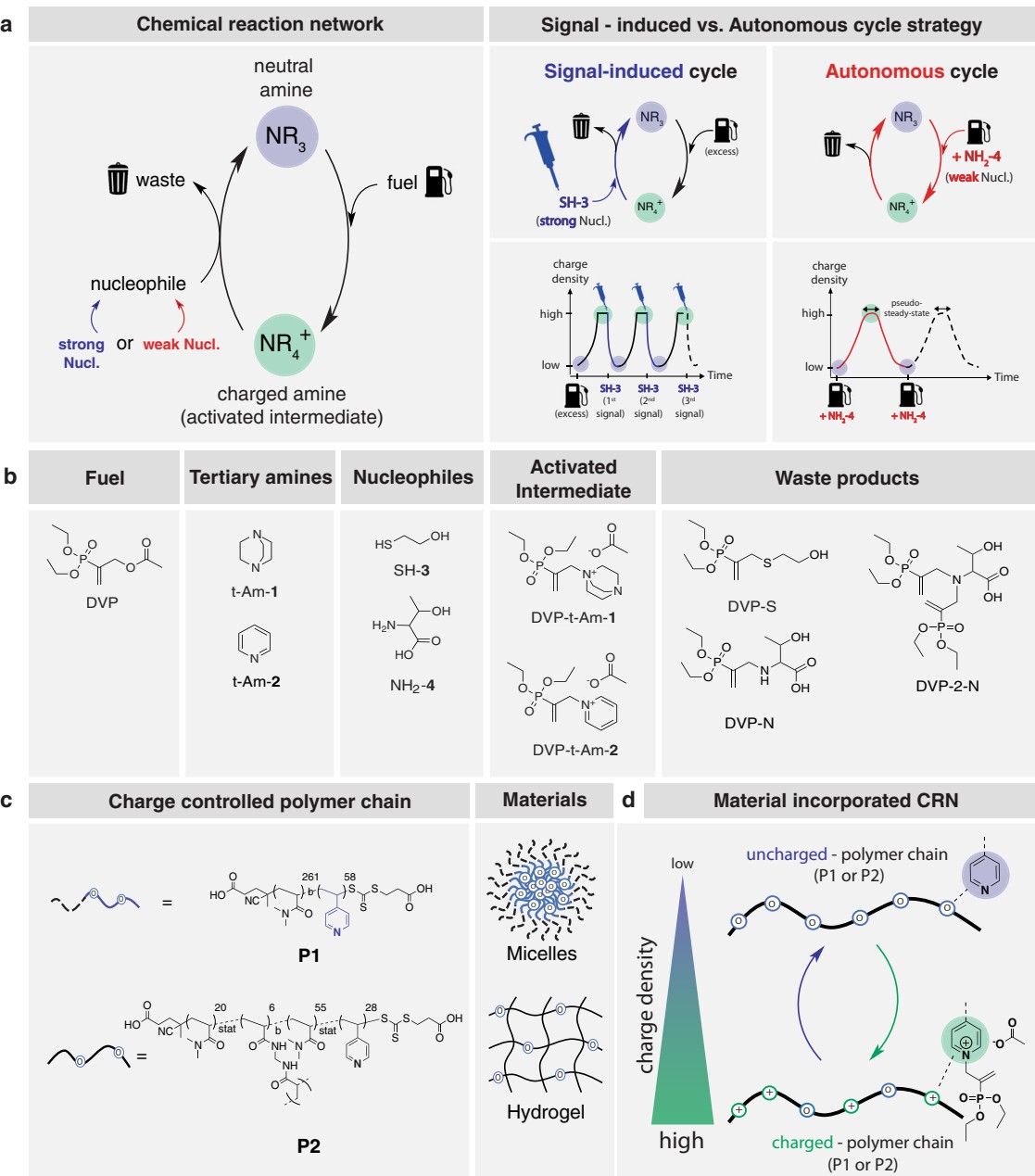

**Fig. 1 | Schematics of the chemical reaction network (CRN), its conditions and components. a** Generic CRN and CRN strategy used to achieve stimuli-responsive or autonomous material changes. **b** Chemical structure of fuel, nucleophiles and tertiary amines (used in the small molecule model CRN), as well as their activated intermediates and waste products. **c** RAFT synthesised polyamine copolymers (P1, P2) used for the preparation of polymeric materials: (1) micellar dispersions and (2) bulk polymer hydrogels. **d** Temporally programmed charge (de)formation on polyamine-functionalized polymer chains using our CRN strategy and charge density distribution. Specifically, fuelled tertiary amines generate charged quaternary nitrogens (activated intermediate), which in turn creates hydrophilic domains along the polymer backbone. A secondary nucleophilic substitution results in the regeneration of the uncharged polyamine by substitution of the intermediate (deactivation) towards the waste product.

Next, we studied systems using the weaker $NH_2$-**4** nucleophile in an effort to achieve transient non-equilibrium ionic species formation (Fig. 2a). First, we explored the reaction rates (Supplementary Fig. 2) and the background reaction (no t-Am-**2**) with a DVP:$NH_2$-**4** ratio of 1:4, which reached full conversion after 46 days (Supplementary Figs. 4 & 5). Introducing t-Am-**2** to the system (t-Am-**2**:DVP:$NH_2$-**4** = 1:2:4), we observed the formation of approximately 69% DVP-t-Am-**2** within the first ~32 h by $^1$H NMR. This was followed by a reaction plateau of ~13 h (pseudo-steady state of the activated intermediate, DVP-t-Am-**2**) and subsequently the recovery of 81% t-Am-**2** in response to the deactivation reaction (decay of DVP-t-Am-**2**, Fig. 2d). When the network

appeared to reach equilibrium (~521 h), an additional 4.0 eq. of $NH_2$-**4** were supplied to the system, which lead to an additional recovery of 12% t-Am-**2** (93% total t-Am-**2** recovery) before equilibrium was re-established.

In an effort to tune the amplitude of the activation reaction, we varied the concentration of DVP fuel. Supply of more fuel (4.0 eq.) accelerated the activation reaction with maximum values of 80% DVP-t-Am-**2**, as shown in Fig. 2e. On the contrary however, the deactivation reaction remained incomplete even with an additional boost of $NH_2$-**4**, levelling off at 92% t-Am-**2** recovery. The extra fuel extended the pseudo-steady-state of the activated intermediate (reaction plateau) to

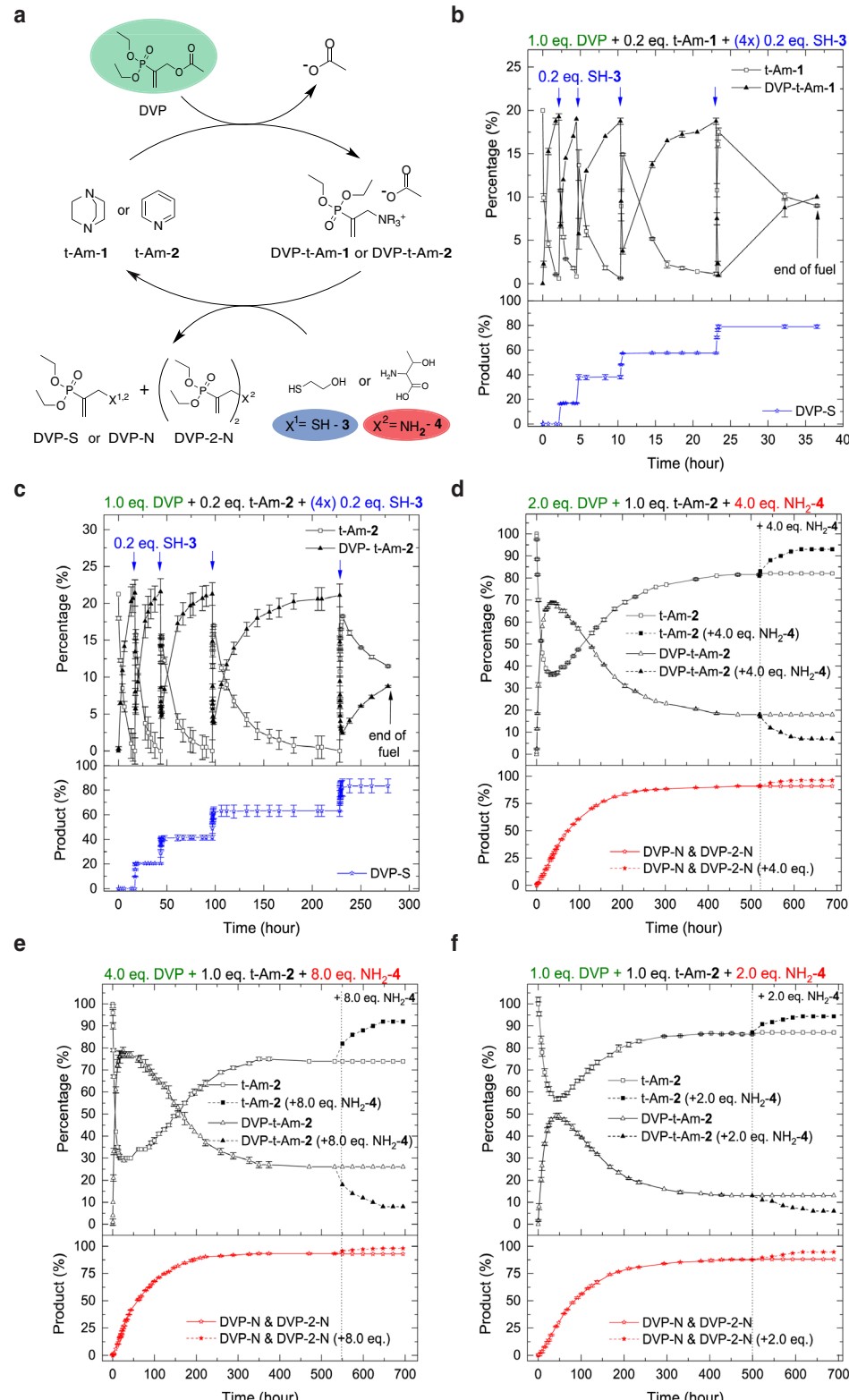

**Fig. 2 | Small molecule CRN model for signal-induced and fuel-driven autonomous cycle. a** CRN of fuel with t-Am-**1** (DABCO) or t-Am-**2** (pyridine) and SH-**3** in signal-induced mode or fuel with t-Am-**2** and excess $NH_2$-**4** in fuel-driven autonomous mode. Conversion of the reactants was monitored by $^1$H NMR over time in $D_2O$/phosphate buffer 1:9 (pH = 7.4, 0.1 M (signal-induced cycle) or 0.5 M (autonomous cycle)) at room temperature. **b** Signal-induced cycle (Supplementary Fig. 6) with t-Am-**1**: DVP (42 mM, 1.0 eq.), t-Am-**1** (0.2 eq.) and four times addition of SH-**3** (0.2 eq.). **c** Signal induced cycle (Supplementary Fig. 7) with t-Am-**2**: DVP (42 mM, 1.0 eq.), t-Am-**2** (0.2 eq.) and four times addition of SH-**3** (0.2 eq.). Autonomous

cycle (Supplementary Figs. 8–10) with fuel/nucleophile variations at constant t-Am-**2** (8.5 mM, 1.0 eq.) concentration (**d**): DVP (2.0 eq.) and $NH_2$-**4** (4.0 eq.), **e** DVP (4.0 eq.) and $NH_2$-**4** (8.0 eq.), **f** DVP (1.0 eq.) and $NH_2$-**4** (2.0 eq.). **d**–**f** At observed equilibrium an additional 4.0, 8.0, and 2.0 equivalents of $NH_2$-**4** were added to one of the duplicate reaction mixtures, respectively. The error bars represent the standard deviation of duplicate measurements. For (**d**, **e**) the product percentages were normalized to 100. Activated intermediates were isolated by exchanging their counterion from acetate to chloride (for full characterisation see Supplementary Figs. 34–43, 54 and 55).

22 h. Evidently, the amount of fuel provides control over the formation of activated intermediate and its reaction plateau. Logically, less fuel (1.0 eq.) slowed down the activation reaction (50% DVP-t-Am-**2** peak formation) and shortened its pseudo-steady-state to ~6 h, while the deactivation reaction remained at a maximum recovery of 94% t-Am-**2**, even after addition of extra NH$_2$-**4** (Fig. 2f).

## Signal-induced micelle disassembly with programmed cargo release and re-uptake

To program the behaviour of a synthetic material, we combined our CRN with micelle forming block-copolymer P1 (Supplementary Figs. 21 & 22), which is based on dimethylacrylamide (DMA) as the water-soluble block and 4-vinylpyridine (4VP) as the hydrophobic block (Fig. 3a). Upon solubilization of 2.2 mg/mL P1 in aqueous-buffer we observed the formation of amphiphilic micelles using dynamic light scattering (DLS) (Supplementary Fig. 13, t = 0 h) and transmission electron microscopy (TEM). TEM revealed uniform, narrowly dispersed micelles (Fig. 3d) with an average diameter (D$_{TEM}$ based on a sample population of $n$ = 1202 in 3.6 μm$^2$) of 18 ± 4.4 nm, as determined by statistical image analysis (Fig. 3c-top). This value agrees with the Z-averaged hydrodynamic diameter (D$_{DLS}$) obtained from DLS

(D$_{DLS}$ = 52 nm, Fig. 3b-bottom). The discrepancy between D$_{TEM}$ and D$_{DLS}$ can be explained by the fact that TEM excludes the length of the hydrophilic DMA chain as a result of dry sample measurements[57]. Having established the formation of micellar dispersions, we first conducted signal-induced micellar (dis)assembly experiments with excess fuel (3.2 eq. DVP), P1 (1.0 eq. 4VP) and consecutive (4x) signal additions (1.0 eq. SH-**3**) (Fig. 3a).

Upon fuelling the micellar solution, the DLS light scatter intensity (scatter count, Fig. 3b-middle) dropped rapidly with a 12 ± 0.7 fold reduction (from 8.1 to 0.6 Mcps) in the first 10 h reaching its minimum at t = 105 h (0.3 Mcps). Simultaneously, $^1$H NMR measurements showed a 28% (t = 10 h) to 74% (t = 105 h) conversion towards the charged activated intermediate (Fig. 3b-top). This observation indicates that partial ionization of the micellar core by DVP (28%) is sufficient to cause a significant switch in the hydrophilicity of the core, leading to micelle disassembly. Hereafter, the disassembled state equilibrium was slowly reached (up to 105 h), due to the decreasing presence of neutral polyamine units. At the same time, amine quaternization on the polymer becomes increasingly dominated by charge repulsion effects[58], which explains the incomplete conversion of DVP (74 ± 4.5%). Micellar solubilization was further confirmed by the measured

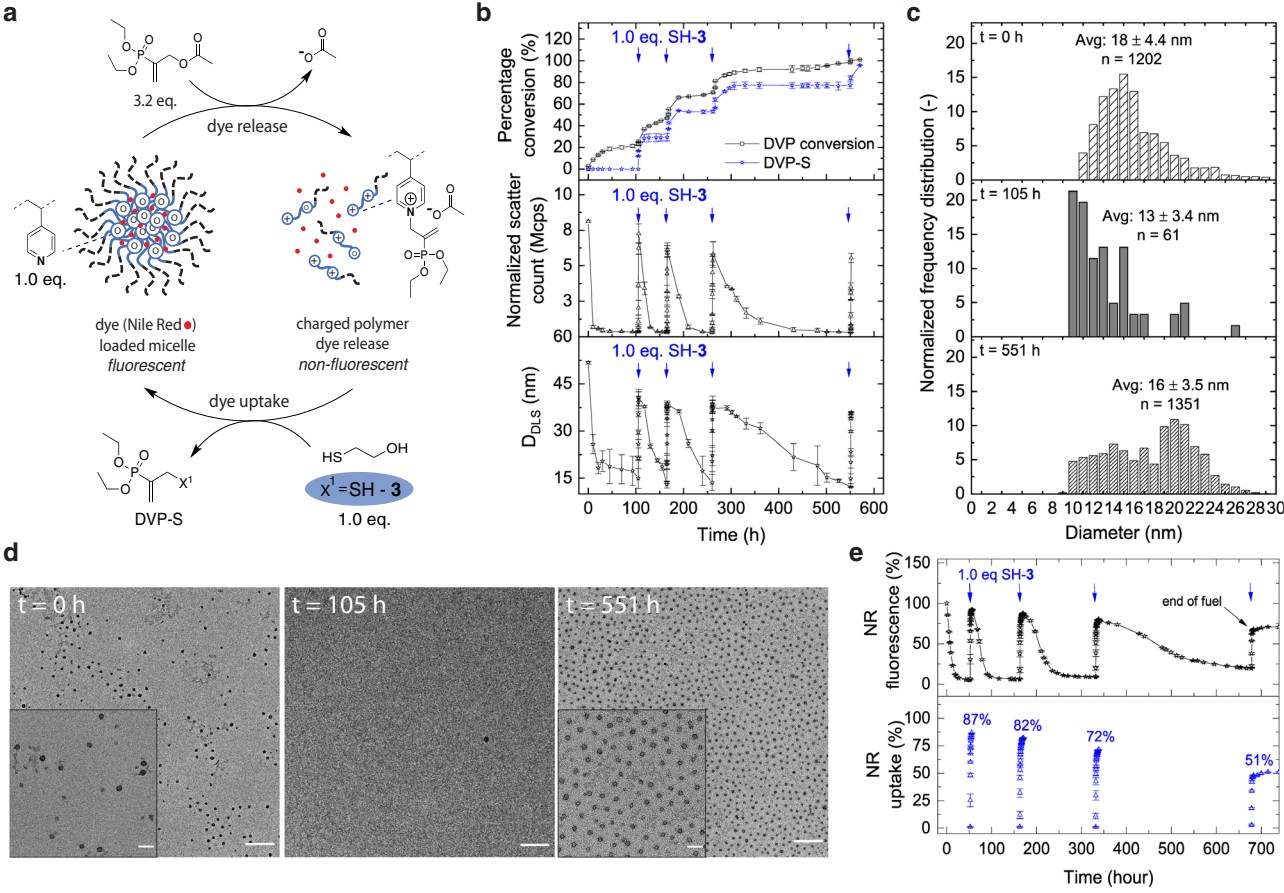

**Fig. 3 | Signal-induced micellar (dis)assembly reaction network in the absence and presence of Nile Red (NR). a** Micellar dispersions (2.2 mg/mL P1) were fuelled with DVP (13 mM, 3.2 eq.) and sequential additions of SH-**3** (1.0 eq.) as signal (blue arrow) in 1.0 mL phosphate buffer (0.1 M, pH = 7.4 and 10% D$_2$O) (Supplementary Fig. 11). Micellar dispersion in (**e**) contained 15.5 ± 0.1 μg NR/mg of polymer (encapsulation efficiency of 39%), while micellar dispersions in (**b**), (**c**) and (**d**) were without NR. **b** Stack graph: (top) DVP conversion and DVP-S formation observed by $^1$H NMR. (middle) Corresponding DLS measured normalized scatter count (Mcps). (bottom) Corresponding DLS measured Z-averaged diameter (D$_{DLS}$ in nm). **c** Normalized frequency distribution based on TEM image analysis for micellar (dis) assembly reaction network. At t = 0 h: initial micelle assembly. At t = 105 h:

disassembled micellar state upon fuel exposure (DVP was added shortly after t = 0 h). At t = 551 h: signal induced micelle (re)assembled state after four sequential cycles and complete consumption of fuel. **d** TEM images of signal-induced micellar (dis)assembly reaction network (Scale bar: 200 nm, insert: 50 nm). (left) At t = 0 h: with no fuel/ signal. (middle) At t = 105 h: deformed micelles upon fuel exposure. (right) At t = 551 h: signal induced micelle (re)assembly after four sequential cycles. **e** Micellar (dis)assembly with corresponding dye uptake profile followed by NR fluorescence at an excitation wavelength of 540 nm and an emission wavelength of 645 nm. The error bars represent the standard deviation of duplicate measurements. Additional DLS and fluorescence data are presented in Supplementary Figs. 13 and 15.

decrease of the $D_{DLS}$ from $52 \pm 0.4$ to $15 \pm 3.2$ nm during the first 105 h (Fig. 3b-bottom), which is consistent with TEM image analysis ($D_{TEM} = 13 \pm 3.4$ nm at t = 105 h, Fig. 3c-middle). In addition, the number of residual micellar structures (Fig. 3d) decreased substantially (n = 61 for an identical image area compared to $n = 1202$ at the start).

To complete the cycle, we initiated the deactivation reaction after 105 h by addition of signal SH-**3** (1.0 eq.). We observed a prompt response in $^1$H NMR and DLS measurements by rapid formation of waste product DVP-S ($93 \pm 8.5\%$ in 1 h), along with an increase in scatter count ($7.2 \pm 0.7$ Mcps) and Z-averaged diameter ($D_{DLS} = 41 \pm 2.5$ nm) to near starting values (Fig. 3b). Importantly, the re-assembled state (at t ~ 106 h) is not at equilibrium, as unreacted excess fuel (-2.2 eq.) spontaneously regenerates the quaternized polyamine units. Subsequently, the cycle starts again, until achieving maximum species ionization leading to micelle disassembly. After confirming three additional consecutive cycles, TEM image analysis (Fig. 3c-bottom) showed the re-occurrence of micellar structures (Fig. 3d) with an average diameter of $16 \pm 3.5$ nm at large population size (n = 1365 in 3.6 $\mu m^2$). Fundamentally, switching between hydrophobic/hydrophilic states by manipulation of the underlying molecular CRN was observed for other micellular systems[13,38,39,59–61] and recently elucidated by mechanistical modelling studies[40]. However, we also observed that the recovered micellar diameter decreased slightly with each cycle, as seen in TEM (from 18 to 16 nm) and DLS (from 51 to 36 nm). Such changes may be caused by the increasing accumulation of waste product with each cycle, which is known to limit cycle efficiency[23,62].

After time-programming micellar (dis)assembly states using SH-signals, we investigated if similar behaviour could be achieved with loaded micelles and whether it is possible to release and re-uptake molecular cargo. To achieve this, micelles were loaded with Nile Red (NR) dye[63] as model cargo. NR is a solvatochromic dye which exhibits strong fluorescence in hydrophobic environments, while in water its fluorescence is quenched[64].

We exposed NR-loaded micelles to fuel (3.2 eq. DVP vs. vinyl pyridine) and monitored their fluorescence. During the first 10 h after fuelling, the fluorescence decreased by $77 \pm 0.3\%$, ultimately reaching a steady $95 \pm 0.1\%$ reduction in fluorescence after 52 h (Fig. 3e). Upon SH-**3** (1.0 eq.) signal addition, the fluorescence increases promptly, reaching a value close to its original intensity, which corresponds to $87 \pm 0.2\%$ NR re-uptake within 4 h (t = 56 h). The transient increase in fluorescence is attributed to the re-established hydrophobic core unit and hence re-assembled micellar structures.

We conducted three additional signal-addition cycling between micelle (dis-)assembly states, which corroborate earlier findings without NR in DLS and $^1$H NMR under identical conditions. Interestingly, we observed an increasingly less efficient cargo re-uptake after each cycle (from $1^{st}$ cycle: 87% to $4^{th}$ cycle: 51%). We experimentally determined that this behaviour is related to the increasing waste accumulation inside the micelle upon signalling (Supplementary Fig. 16), as seen in micellar fuelling experiments without NR.

### Autonomous micelle (dis)assembly with programmed cargo release – uptake

Having established controllable signal-responsive micellar (dis) assembly, we further investigated autonomous cycling by concurrent competition between activation and deactivation reactions in the CRN with an excess of NH$_2$-**4**. Autonomous fuelling experiments were performed at optimized conditions with 2.2 mg/mL P1 (which corresponds to a 4VP concentration of 4.0 mM), 2.0 eq. DVP and 8.0 eq. NH$_2$-**4** in aqueous-buffer (Fig. 4a). The micelle disassembly process reaches its apex at t = ~48 h, corresponding to $59 \pm 2.0\%$ formation of charged pyridine units (activated intermediate) after 1.1 eq. DVP conversion ($55.5 \pm 0.9\%$) and 0.52 eq. of waste product formation ($26 \pm 0.2\%$ DVP-N and DVP-2-N) (Fig. 4b-top). We found that the transition to the disassembled state occurred upon formation of ~60% of

activated intermediate (t = 48 h), as demonstrated by a simultaneous drop in DLS measured scatter count from 11.9 to 0.5 Mcps and $D_{DLS}$ from 53 to 23 nm (Fig. 4b-middle/bottom). Notably, we observe a larger number of remaining micellar structures (t = 48 h) in contrast to the signal-induced micellar (dis)assembly, in parallel with a $D_{TEM}$ change from $19 \pm 4.5$ nm (n = 1065) before fuelling (t = 0 h) to $15 \pm 4.2$ nm (n = 91) after 48 h (Fig. 4c, d).

Nevertheless, from 48 to 504 h the deactivation reaction kinetically outperforms the activation reaction due to continued depletion of fuel reserves (from 55.5 to -100%). This ultimately leads to further accumulation of waste product (from 26 to 95%) until equilibrium is reached after approximately 504 h (Fig. 4b-top). During that period, DLS measured scatter count increases from 0.5 to 11.4 Mcps, while $D_{DLS}$ recovered to 42 nm (Fig. 4b-middle/bottom). In good agreement with DLS, TEM image analysis also revealed an increase in micellar diameter ($D_{TEM} = 17 \pm 2.6$ nm) and qualitatively a much larger number of micellar structures could be observed (n = 706) (Fig. 4c, d). As the CRN depletes its fuel reserve, the rate of deactivation becomes higher than the activation rate and thus reverts the charged polyamine units to their uncharged precursor. This leads to increasing hydrophobicity on the polymer backbone, which ultimately results in reassembled micellar structures at equilibrium state.

Lastly, we addressed autonomous cargo release and re-uptake by conducting NR probed micellar fuelling experiments at conditions identical to the no-cargo micellar experiments (Fig. 4e). Using the fuel-driven out-of-equilibrium CRN strategy, $83 \pm 0.3\%$ of cargo was released in 55 h. Hereafter, fluorescence intensity steadily increased back to nearly the original level, corresponding to $80 \pm 1.0\%$ cargo re-uptake over the next ~19 days. This way, we achieved autonomous (dis) assembly of micellar-macromolecular structures with time-programmed cargo release and re-uptake.

### Temporally programmed hydrogel swelling

To further demonstrate our CRN strategy, we aimed to achieve reversible expansion–contraction of a polymeric hydrogel network by temporally controlling its water in – and outflux (Fig. 5a). We co-polymerized DMA and 4VP to generate a water-soluble statistical copolymer (P2 precursor - Supplementary Fig. 24). We then cross-linked this copolymer with bis(acrylamide) to form cube shaped polymer hydrogels (P2) with polyamine concentrations of 97 mg/mL. The resulting gels had a water content of 90 wt% and dimensions of approximately $1.4 \times 1.2 \times 0.5$ cm (L/W/H). We hypothesized that network expansion can be induced by charge generation upon fuel addition. The increasing concentration of charges in the polymer network will lead to an increase in osmotic pressure in the material which makes the hydrogel swell (expansion) until it is balanced by elastic network forces[54,65]. In addition, potential repulsive forces between the ionized activated intermediates on the polymer might add to the swelling forces and thus to the expansion of the hydrogel[66].

To test this hypothesis, we observed fuelled/non-fuelled hydrogels via photographs over time and evaluated their swelling percentage (S%, Eq. 5), where the difference in weight between equilibrium swollen hydrogels (at t = 0) and the fuel activated swollen hydrogels (t) is compared. First, polyamine-containing hydrogels were activated with DVP (43.2 mM, 1.0 eq. vs. vinyl pyridine). Successful material expansion (Fig. 5b) was observed with a maximum of $106 \pm 16\%$ increase in S% 96 h after fuelling (Fig. 5d-blue), while control hydrogels remained at equilibrium weight (Fig. 5d-grey). This drastic increase in volume was further confirmed visually by comparing fuelled hydrogels to their non-fuelled counterparts (Fig. 5b). Next, we investigated the reversibility of the system by addition of SH-**3** (1.0 eq.). The nucleophilic triggering of the system resulted in hydrogel contraction to near starting values 96 h after signal introduction (S% = $7 \pm 8\%$). To demonstrate the repeatability of the system, signal-responsive (de) swelling was repeated by re-fuelling (S% = $104 \pm 22\%$) and subsequent

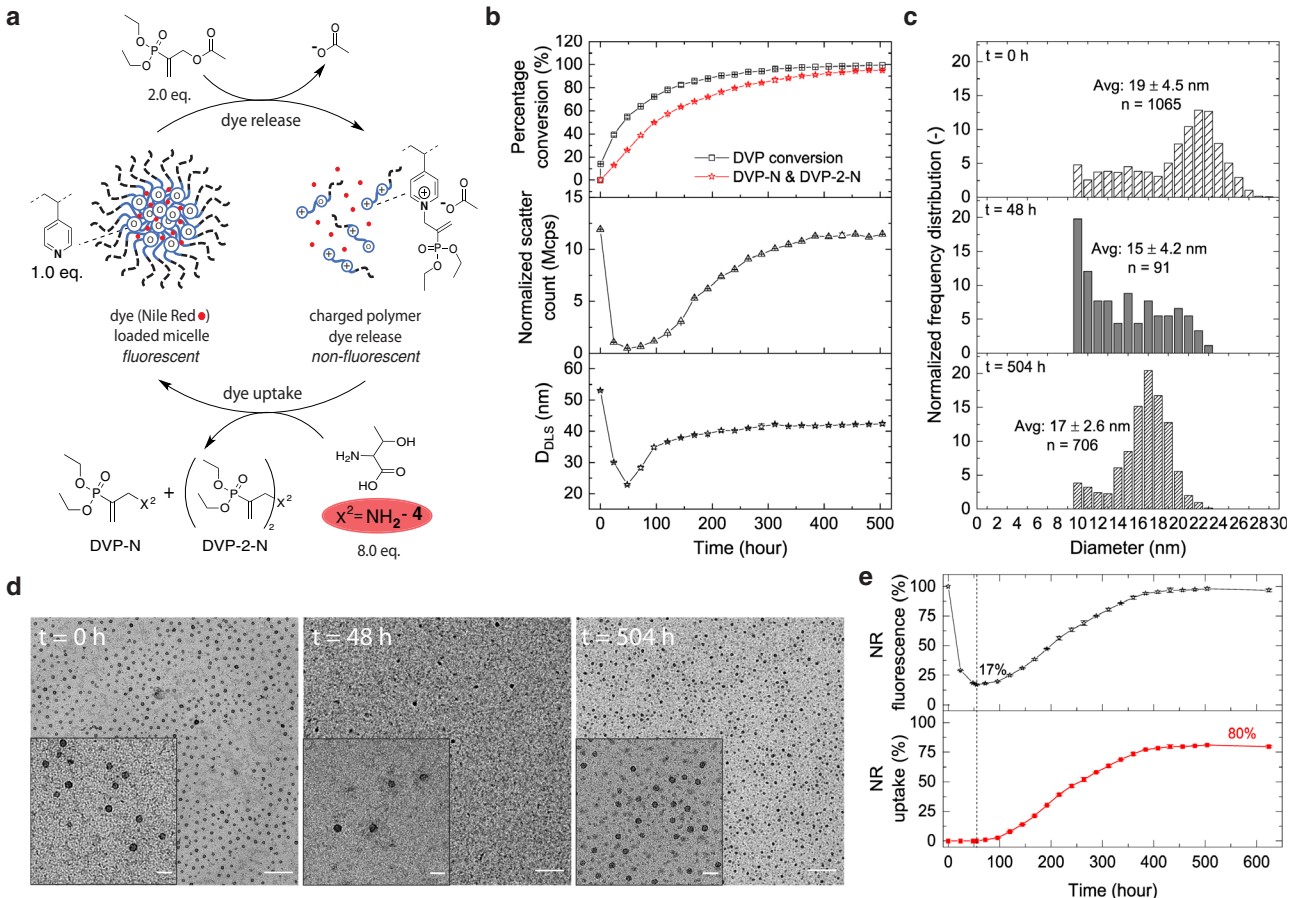

**Fig. 4 | Fuel-driven out-of-equilibrium micellar (dis)assembly reaction network in the absence and presence of Nile Red (NR). a** Micellar dispersions (2.2 mg/mL P1) were fuelled with DVP (8.0 mM, 2.0 eq.) and NH$_2$-**4** (8.0 eq.) in 1.0 mL phosphate buffer (0.5 M, pH = 7.4 and 10% D$_2$O) (Supplementary Fig. 12). Micellar dispersion in (**e**) contained 15.5 ± 0.1 μg NR/mg of polymer (encapsulation efficiency of 39%), while micellar dispersions in (**b**), (**c**) and (**d**) were without NR. **b** Stack graph (top) DVP conversion and DVP-N & DVP-2-N formation observed by $^1$H NMR. (middle) Corresponding DLS measured normalized scatter count (Mcps). (bottom) Corresponding DLS measured Z-averaged diameter (D$_{DLS}$ in nm). **c** Normalized frequency distribution based on TEM image analysis for micellar (dis)assembly reaction network. At t = 0 h: initial micelle assembly. At t = 48 h: disassembled

micellar state upon fuel exposure (DVP was added shortly after t = 0 h). At t = 504 h: micellar (re)assembled state. **d** TEM images of fuel-driven out-of-equilibrium micellar (dis)assembly reaction network (Scale bar: 200 nm, insert: 50 nm). (left) At t = 0 h: with no fuel/ nucleophile. (middle) At t = 48 h: micellar disassembled state upon fuel/nucleophile exposure. (right) At t = 504 h: micellar (re)assembled state. **e** Micellar (dis)assembly with corresponding dye uptake profile followed by NR fluorescence at an excitation wavelength of 540 nm and an emission wavelength of 645 nm. The error bars represent the standard deviation of duplicate measurements. For (**b**–top) DVP conversion and product percentages were normalized to 100. Additional DLS and fluorescence data are presented in Supplementary Figs. 14 and 15.

SH-**3** signalling (S% = 6 ± 10%) to temporally program the materials swelling behaviour (Fig. 5b, d-blue). Turning our attention to kinetics, we were surprised that swelling proceeds at a similar time-scale as de-swelling (t ~96 h). We suggest here, that the large hydrogel size limits SH-**3** diffusion into the material. Another potential factor, which affects the diffusive influx of SH-**3**, is the convective counterforce of water out-flux with decreasing ionization.

Next, we explored transient hydrogel swelling by exposing equilibrium-swollen gels to DVP (13.2 mM, 2.1 eq. vs. vinyl pyridine) and excess NH$_2$-**4** (8.0 eq.) (Fig. 5a). We observed temporary swelling of the polymeric hydrogel network with clear swelling maxima around t = 168 h (S% = 80 ± 11%), followed by de-swelling which approached starting values (S% = 4 ± 6%) after 504 h (Fig. 5c, d-red). This transient swelling study on synthetic hydrogel materials in aqueous media has shown a robust material response and thus the versatility of using the programmed solvent-material interaction strategy for autonomous cycling.

In this work, we have introduced a new CRN which operates through successive nucleophilic substitutions on electron deficient allyl acetates (fuel). By first combining a tertiary nitrogen species with the fuel, a cationic quaternary nitrogen intermediate can be formed.

This intermediate is stable in pH 7.4 aqueous solutions until undergoing a second nucleophilic substitution with nucleophiles such as a thiol or primary amine. This process regenerates the starting neutral tertiary amine species, along with the formation of a waste product. Unlike most chemically fuelled non-enzymatic CRNs, the deactivation reaction can be controlled at constant pH by judicious choice of nucleophile. With strong nucleophiles such as 2-mercaptoethanol we were able to achieve signal-induced cycling between charge states with excess fuel (DVP) and sequential additions of nucleophile. By switching to weaker nucleophiles such as threonine, both fuel and nucleophile can be introduced simultaneously to yield fuel-driven out-of-equilibrium or autonomous cycling. By incorporating the tertiary nitrogen species into polymeric scaffolds, cycling between neutral and cationic amine species yields transitions between collapsed, hydrophobic polymer chains and solvated, hydrophilic polymer chains. In micellar dispersions, this allowed for the programmed release and re-uptake over time of a solvatochromic dye, acting as a model cargo. In bulk polymer hydrogels we used the CRN to control water influx, allowing us to control hydrogel swelling behaviour by fuel and nucleophile additions. Both material classes could operate under signal-induced and autonomous cycling conditions, leading to

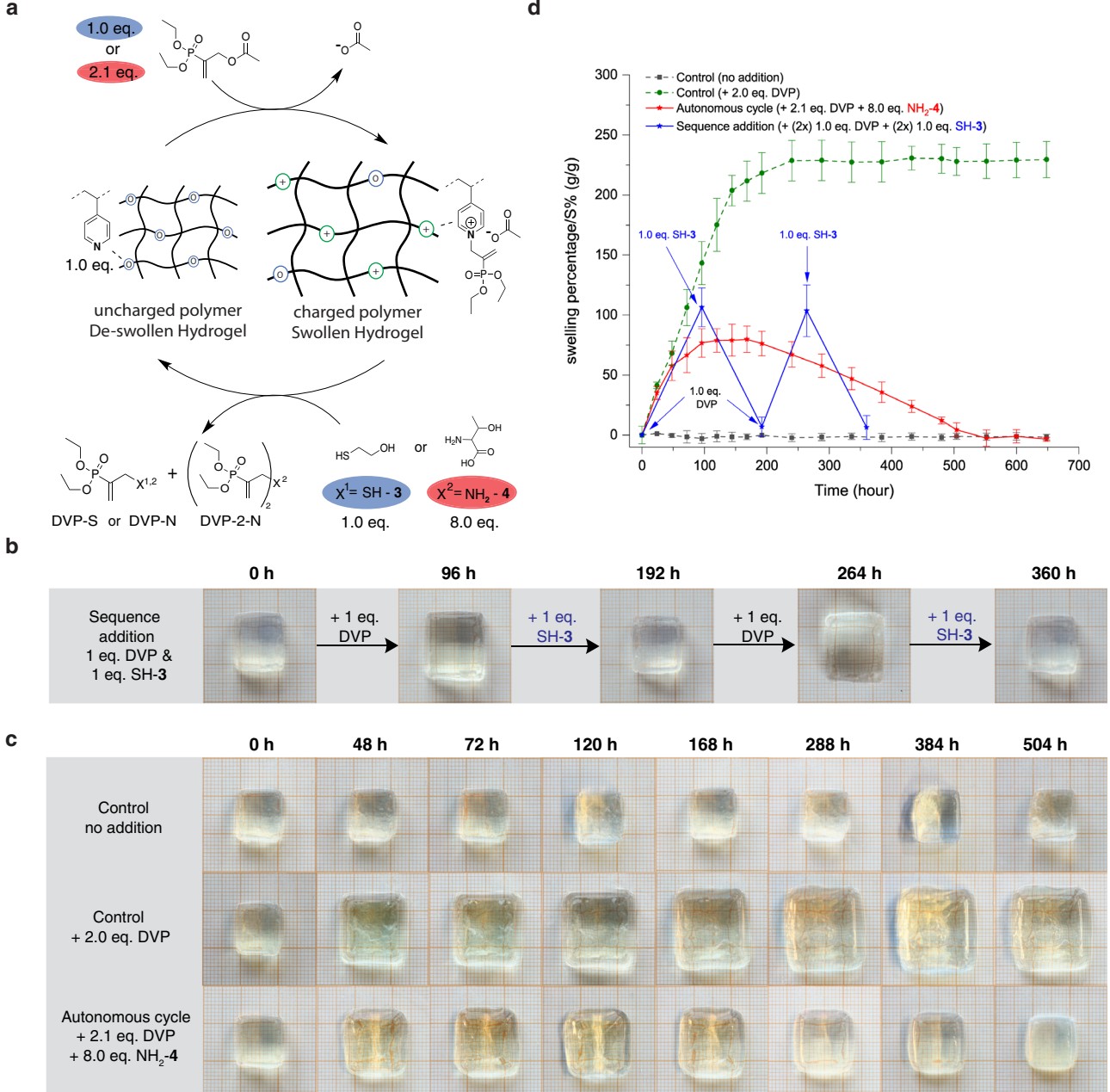

**Fig. 5 | Temporally programmed hydrogel swelling and de-swelling. a** Hydrogel (de)swelling using signal-responsive or autonomous CRN strategy. The transition from un-swollen to swollen state was monitored visually by time-lapse photographs on millimetre paper and via hydrogel weight measurements (run in triplet) over time in phosphate buffer (pH = 7.4, 0.1 M) at room temperature. **b** Time-lapse photographs of signal-responsive hydrogels without solution. **c** Time-lapse photographs of fuel-driven out-of-equilibrium cycle hydrogels without solution in comparison to control hydrogels. **d** Swelling percentage (S%) for temporally programmed signal-responsive (blue) and autonomous (red) hydrogel (de)swelling over time. Control hydrogel measurements without reactants (grey) and with 2.0 eq. DVP (green). The error bars represent the standard deviation. Additional hydrogel time-lapse photographs are shown in Supplementary Figs. 17–19.

different types of behaviour. The principle of temporally programming the behaviour of synthetic materials, shown in this work, is applicable not only to constructing out-of-equilibrium synthetic structures but also as advanced strategy for controlled molecular cargo delivery. By using the underlying CRN, its scalability and applicability to different material classes combined with its possibilities of tuning the kinetic profile based on choice of nucleophile or choice of fuel[67], a variety of new designs for interactive structures can be envisioned. We therefore anticipate that this concept will contribute to the development of next generation soft materials, where signal or time-programmed control over charge density allows for interactive and adaptive material properties, such as stiffening, adhesion or motion.

## Methods

### NMR spectroscopy

We recorded NMR spectra on an Agilent-400 MR DD2 NMR instrument at 25 °C (399.7 MHz for ${}^1$H, 100.5 MHz for ${}^{13}$C and 161.9 MHz for ${}^{31}$P) using residual solvent signals as internal reference. To suppress the water peak, we used the PRESAT configuration (suppress one highest peak). MNova NMR software (Mestrelab Research) was used to process NMR spectra. The polymerization conversion ($\rho$) was calculated by monitoring reduction in the ${}^1$H NMR integrals of the unsaturated protons of the monomer (∫M: 5.6 – 6.7 ppm for DMA, 5.5 – 6.7 ppm for 4VP) and aromatic protons in case of 4VP (7.5 ppm) relative to the internal standard DSS (~0 ppm). In the case of a copolymerization with

both DMA and 4VP, we calculated the conversion of both monomers via Eq. 1.

$$\rho = \frac{\int M(t0) - \int M(t)}{\int M(t0)} \qquad (1)$$

For a polymerization containing z monomers, $M_{n,conv}$ was calculated using to Eq. 2. In Eq. 2, $[Mx]_0$ is the initial concentration of monomer x, $[CTA]_0$ is the initial chain transfer agent (CTA) concentration and $M_{Mx}$ and $M_{CTA}$ are the monomer x and CTA molecular weights, respectively.

$$M_{n,conv} = \sum_{x=1}^{Z} \rho * \frac{[M]_0}{[CTA]_0} * M_{Mx} + M_{CTA} \qquad (2)$$

## Micellar dispersions and DLS measurements

The signal-induced cycle was performed with P1 (2.24 mg, 4.0 µmol, 1.0 eq. 4VP), DVP (3.2 eq.) in buffer and addition (4x) of the signal SH-3 (1.0 eq.). Stock solutions of P1 and DVP were mixed in a 4.0 mL vial (total reaction volume 1.0 mL), stirred vigorously for 10 s and immediately transferred to PMMA disposable cuvettes and measured at specific time points. The signal SH-3 (0.28 µL, 4.0 µmol) in buffer was added to the cuvette, 4 times at specific time points. In contrast, for the autonomous cycle P1 (4.0 µmol, 1.0 eq. 4VP), DVP (2.0 eq.) and NH₂-4 (8.0 eq.) in buffer were mixed together in a 4.0 mL vial (total reaction volume 1.0 mL), stirred vigorously for 10 s and immediately transferred to PMMA disposable cuvettes and measured for t = 0 h and then every 24 h.

## TEM measurements

The morphologies of the micelles and their size were observed via Transmission Electron Microscopy (TEM). For both, the signal-induced and autonomous cycle, aliquots were taken at specific time points from the DLS measurement samples and drop casted onto a Formvar/Carbon 400 mesh Cu grid. The grids were exposed (30 s) to 3.0 µL uranyl acetate stain solution (2 wt% in H₂O), then washed (3x) with milli-Q water and dried on filter paper before loading it on the TEM single tilt holder. TEM pictures were processed and analysed using ImageJ to obtain the particle size distribution.

## GPC measurements

We measured the average molecular weight and dispersity (Đ) of the synthesized polymers using a Shimadzu GPC with DMF containing LiBr (25 mM) as eluent. The GPC system consists of a Shimadzu CTO-20AC Column oven, a Shimadzu RID-10A refractive index detector, a Shimadzu SPD-20A UV-Vis detector, PL gel guard column (MIXED, 5 µm), 50 mm × 7.5 mm, and 1 × Agilent PLGel (MIXED, 5 µm), 300 mm × 7.5 mm, providing an effective molar mass range of 200 to 2 × 10⁶ g/mol. The eluent was used at 50 °C with a flow rate of 1.0 mL/min. We used low dispersity PMMA standards (Sigma Aldrich) ranging from 800 to 2.2 × 10⁶ g/mol to calibrate the GPC columns, and molar masses are reported as PMMA equivalents. The log $M_P$ vs. time calibration curve was fit to a 3rd-order polynomial for both systems, which was near linear across the molar mass ranges.

## Fluorescence measurements and micelle loading

A Nile Red solution in THF (20.0 µL, 2.0 mg/mL) was added to a micellar dispersion P1 (2.0 mL, 12.0 µmol, 1.0 eq. 4VP) in buffer and incubated in the dark in an open vial to evaporate overnight the organic solvent (the mixture was measured in the NMR confirming that all THF had been evaporated before further usage). This solution was then mixed with DVP (3.2 eq.) in buffer to a total reaction volume of 3.0 mL and transferred to quartz cuvettes (path length of 1 cm) and immediately measured (t = 0 h). The signal SH-3 (0.84 µL, 12.0 µmol, 1.0 eq.) was added in the cuvette at specific time points, which was

repeated three times, hereafter (at specific time points). In case of the autonomous cycle, a Nile Red treated micellar dispersion (2.0 mL, 12.0 µmol, 1.0 eq. 4VP) was mixed with DVP (2.0 eq.) and NH₂-4 (8.0 eq.) in buffer and transferred in quartz cuvettes (total reaction volume 3.0 mL) and immediately measured (t = 0 h). Hereafter, the reaction mixture was measured every 24 h. Each solution was measured in the Fluorometer at an excitation wavelength of 540 nm and an emission wavelength of 645 nm. To determine the loading of the micelles, a micellar dispersion (2.0 mL, 12.0 µmol) was prepared with Nile Red (20 µL, 2.0 mg/mL in THF) addition and overnight solvent evaporation in the dark. The samples were centrifuged at 5000 rpm for 10 min and ~900 µL buffer was removed carefully without disturbing the pellet and replaced with fresh phosphate buffer, this step was repeated three times. After another centrifugation step and removal of the buffer, 900 µL of DMF was added and the vial was shaken to dissolve the loaded micelles. The fluorescence of the solution was measured at an excitation wavelength of 540 ± 20 nm and emission wavelength 620 ± 30 nm and compared to the calibration curve of known concentrations of Nile Red in 90% DMF in buffer (Supplementary Fig. 1), to determine the Nile Red loading per mg of polymer P1.

## Reaction rate constants comparison: Reaction of DVP with t-AM-1 & 2 and nucleophiles

Briefly, DVP (10.0 mg, 42 µmol, 1.0 eq.) and DSS as internal standard (1.0 eq.) were dissolved in 0.1 mL D₂O/0.4 mL phosphate buffer (0.1 M, pH = 7.4). Then, either t-Am-1, t-Am-2, SH-3 or NH₂-4 (0.2 eq.) dissolved in 0.5 mL phosphate buffer (0.1 M, pH = 7.4) were added to the reaction mixture. The reactions were immediately followed by ¹H NMR.

## Fitting pseudo-first order reaction kinetics

To compare the reaction between DVP with t-AM-1/t-AM-2 and DVP with SH-3/NH₂-4, reactions were performed at pseudo-first order conditions by using one of the reactants in excess (DVP). The concentrations used were 0.04 M (1.0 eq.) of DVP with 0.008 M (0.2 eq.) of either tertiary amines or S,N -terminal nucleophiles. The pseudo- first order reaction rate constants were calculated by fitting the conversion of DVP over time with Eq. 3:

$$ln\left(\frac{[B]_t}{[B]_0}\right) = -k[A]_0 t \qquad (3)$$

where $[B]_0$ is the initial concentration of DVP in excess at t = 0, 0.04 M; $[B]_t$ is the concentration of DVP at every specific time point obtained by ¹H NMR, with DSS as internal standard; k is the pseudo-first order reaction rate constant and $[A]_0$ is the concentration of reactant (SH-3, NH₂-4, t-AM-1 or t-AM-2 = 0.008 M).

## Reaction of DVP with S,N - terminal nucleophiles (Blank reactions)

**Reaction of DVP with SH-3**. DVP (10.0 mg, 42 µmol, 1.0 eq.) and DSS as internal standard (1.0 eq.) were dissolved in 0.1 mL D₂O/0.4 mL phosphate buffer (0.1 M, pH = 7.4). Then, SH-3 (1.0 eq.) dissolved in 0.5 mL phosphate buffer (0.1 M, pH = 7.4) was added to the reaction mixture. The reaction was immediately followed by NMR until completion at t = 110 h.

**Reaction of DVP with NH₂-4**. DVP (4.0 mg, 17 µmol, 1.0 eq.) and DSS as internal standard (1.0 eq.) were dissolved in 0.1 mL D₂O/0.4 mL phosphate buffer (0.5 M, pH = 7.4). Then, NH₂-4 (4.0 eq.) dissolved in 0.5 mL phosphate buffer (0.5 M, pH = 7.4) was added to the reaction mixture. The reaction was immediately followed by NMR until completion at t = 1104 h.

**Thiol-addition conditions for signal-induced cycle – molecular study**. DVP (10.0 mg, 42 µmol, 1.0 eq.) and DSS as internal standard (1.0

**Table 1 | Autonomous cycle reactant data**

| DVP (mg, mM) | DSS (mM) | t-Am-2 (mM) | NH₂-4 (mM) | DVP:t-Am-2:NH₂-4 (eq.:eq.:eq.) | Added NH₂-4 at Equilibrium (eq.) |
|---|---|---|---|---|---|
| 2.0, 8.5 | 8.5 | 8.5 | 17.0 | 1.0:1.0:2.0 | 2.0 |
| 4.0, 17.0 | 17.0 | | 34.0 | 2.0:1.0:4.0 | 4.0 |
| 8.0, 33.8 | 33.8 | | 68.0 | 4.0:1.0:8.0 | 8.0 |

Briefly, DVP, DSS as internal standard, t-Am-**2** and NH₂-**4** were dissolved in 0.1 mL D₂O/0.9 mL phosphate buffer 1:9 (0.5 M, pH = 7.4) and immediately followed by ¹H NMR. Upon observed reaction equilibrium, additional NH₂-**4** (2.0 eq., 4.0 eq. and 8.0 eq.) was added to the corresponding reaction mixture to drive the reaction to completion.

eq.) were dissolved in 0.1 mL D₂O/0.4 mL phosphate buffer (0.1 M, pH = 7.4). Then, t-Am-**1** or t-Am-**2** (0.2 eq.) dissolved in 0.5 mL phosphate buffer (0.1 M, pH = 7.4) were added to the reaction mixture to a total concentration of 8.5 mM of t-Am-**1** or t-Am-**2**. The reaction was immediately followed by ¹H NMR. Upon complete formation of intermediate (DVP–t-Am-**1** or DVP–t-Am-**2**), SH-**3** (0.2 eq.) in buffer was added to the solution and followed by ¹H NMR. Hereafter, sequential addition of thiol-signal was repeated three times until all DVP had been consumed.

**Amine-addition conditions for autonomous cycle – molecular study.** The autonomous cycle reaction was subject to three conditions in which the DVP concentration was varied from 1.0 eq. to 4.0 eq. and NH₂-**2** from 2.0 eq. to 16.0 eq. at constant t-Am-**2** concentration (1.0 eq.), as follows in Table 1.

**Micelle assembly - disassembly experiments.** The signal-induced/ autonomous cycle experiments for micelle assembly and disassembly were performed with reactant concentrations as described in section: Micellar dispersions and DLS measurements. After addition of the reactants, the reaction mixture was vigorously stirred for 10 seconds and immediately followed by ¹H NMR.

**Hydrogel preparation (P2).** p(4VP₂₈-stat-DMA₅₅) (292.0 mg, 33.3 µmol) was mixed together with BisAM (31.0 mg, 200 µmol) and DMA (99.0 mg, 1000 µmol) in 2.5 mL deionized water and degassed for 10 min under Argon. Separately, stock solutions of TEMED and APS were prepared in 0.5 mL deionized water and degassed for 10 min under Argon. TEMED (266.6 mM) and APS (133.3 mM) were added to the degassed p(4VP₂₈-stat-DMA₅₅), BisAM, DMA mixture and shaken for 1 min before adding the solution into a cast (0.7 mL per cast) and leaving it for curing overnight. The crude hydrogels were removed from their casts and washed twice with 10 mL phosphate buffer (0.1 M, pH = 7.4) and then left for 96 h (the supernatant was exchanged with buffer every day) until equilibrium swelling had been achieved, before further usage.

**Hydrogel swelling - de-swelling experiments.** Hydrogels were swollen in buffer (0.1 M, pH = 7.4) for 96 h until equilibrium weight had been achieved. The equilibrium swelling percentage (equilibrium S%) was found by dividing the equilibrium swollen polymer $W_s$, by the weight of the oven dried polymer $W_d$, according to Eq. 4:

$$equilibrium\ S\% = \frac{W_s - W_d}{W_d} * 100 \qquad (4)$$

The avg. equilibrium swelling percentage (equilibrium S%) was 1975 ± 165%, for all hydrogels used in this work. Water uptake and water loss, was calculated based on the weight difference between the equilibrium swollen hydrogels ($W_{equil.}$) and the time-observed hydrogels ($W_t$). The S% (t) is the S% at each time point (t) minus the S% at t = 0

after equilibrium swelling had been achieved (Eq. 5). Hence, each hydrogel is normalized in reference to its equilibrium swollen state (t = 0).

$$S\%(t) = \left(\frac{W_t - W_{equil.}}{W_{equil.}} * 100\right)_t - \left(\frac{W_{t=0} - W_{equil.}}{W_{equil.}} * 100\right)_{t=0} \qquad (5)$$

Hydrogels without added reactants (blank hydrogels) were submerged in 10 mL buffer. Their weight and photograph were taken at specific time points as stated for the hydrogels with added reactants. The non-fuelled hydrogels remained around starting values (Avg. S% = −6 ± 1.4%) for the entire observation time. The control with 2.0 eq. DVP, reached equilibrium after 240 h (Avg. S% = 187 ± 2.5%, from 240 h to 648 h) (Supplementary Fig. 17).

**Signal-induced hydrogel swelling and de-swelling.** Equilibrium swollen hydrogels were placed in petri-dishes, weighed on a balance and photographed on top of millimetre paper (t = 0). Then, a stock solution of DVP (203.9 mg, 0.863 mmol) was prepared in 13.5 mL buffer (0.1 M, pH = 7.4). To each hydrogel (n = 3) 4.2 mL of DVP stock solution (0.22 mmol, 1.0 eq.) was added (total reaction volume per hydrogel 5.2 mL). Time lapse photographs (Supplementary Fig. 18) were taken every 24 h until no further swelling was observed (t = 96 h). Next, the remaining solution was removed using a pipette and stored. Hydrogels were dried, if necessary with paper tissue, before the weights of the swollen hydrogels were taken ($W_t$). After re-introduction of the stored solution, SH-**3** (1.0 eq.) prepared in buffer was added to the DVP-swollen hydrogels and again followed by taking every 24-hour photographs until no further de-swelling was observed (t = 192 h). Hydrogels were then weighed without solution and photographed. The addition of DVP (1.0 eq.) for hydrogel swelling and the addition of SH-**3** (1.0 eq.) for de-swelling was repeated hereafter on the same hydrogels by using fresh stock solutions.

**Autonomous hydrogel swelling and de-swelling.** The conditions used in this experiment deviate from the previous procedure, since we optimized the reaction buffer and sterilized the hydrogels with sodium azide (0.02 wt%) against microbial growth before fuelling. Indeed, the availability of phosphates from buffer and a nitrogen source from amino acids provided favourable conditions for biological growth. Since the fuel reacts with sodium azide[47], we supplied extra fuel (additional 0.1 eq.) to the system, keeping in mind that sufficient DVP will be available (2.0 eq.) for the transient cycle (2.1 eq. DVP). Phosphate buffer at 0.1 M was chosen because of the good solvation properties of the material compared to 0.5 M, in which the polymers collapsed. Due to the decrease in buffer strength, however, we diluted the system to sustain consistent pH conditions. Equilibrium swollen hydrogels were placed in petri-dishes, weighted on a balance and photographed on top of millimetre paper (t = 0). Hereafter, a stock solution of NaN₃ (9.0 mg, 0.138 mmol) was prepared in 3.0 mL buffer (0.1 M, pH = 7.4, 1:9). To each hydrogel (n = 3), 1.0 mL of stock solution was added and the hydrogels were left for additional 24 h. Next, stock solutions of DVP (326.7 mg, 1.38 mmol) in 30 mL buffer and NH₂-**4** (627.7 mg, 5.27 mmol) in 30 mL buffer were prepared. Three vials were then mixed with stock solutions (ratio NH₂-**4**:DVP:buffer = 10 mL:10 mL:13 mL) and shaken for 1 min before being added to the three hydrogels (total reaction volume per hydrogel = 35 mL). Time lapse photographs were taken every 24 h for two hydrogels without solution and two hydrogels with solution. Hydrogel weights ($W_t$) were taken from two hydrogels every 24 h. Note, that one hydrogel was left 'undisturbed' in the reaction mixture, meaning that the reaction solution was not removed for weighing purposes (Supplementary Fig. 19). This hydrogel was only photographed and compared to the 'disturbed' variants, for which the reaction mixture was removed for weighing and hereafter re-supplied.

## Data availability

All data generated and analysed during this study are included in this article and its Supplementary Information/Source data files. The data used in this study are available in the 4TU.ResearchData database [https://doi.org/10.4121/21197290.v1][68]. Source data are provided with this paper.

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

## Acknowledgements

The authors acknowledge financial support by the European Research Council (ERC Consolidator Grant 726381) to R.E., B.K., and R.W.L. and by the Netherlands Organisation for Scientific Research and the National Natural Science Foundation of China (NWO-NSFC joint project) to I.P., We thank Dr. G.A. Filonenko for help with the distillation setup, NMR sample preparation and valuable discussions.

## Author contributions

R.E. conceived and designed the research. B.K. designed and carried out the experiments and analysed the results, I.P. performed the fluores-cence measurements and assisted with NMR analysis. R.W.L. synthe-sised the polymers and assisted with TEM measurements. B.K. wrote the manuscript. I.P., R.W.L. and R.E. revised the manuscript. All authors commented on the work and the manuscript.

## Competing interests

The authors declare no competing interests.
