## [Peer Review File · Nature Communications]

Temporally programmed polymer – solvent interactions using a chemical reaction networkEditorial Note: This manuscript has been previously reviewed at another journal that is not operating a transparent peer review scheme. This document only contains reviewer comments and rebuttal letters for versions considered at Nature Communications.

Reviewers' Comments:

Reviewer #2:

Remarks to the Author:

The authors have addressed most of the comments properly. However, since the answer letter was cut to a few remaining author-selected parts, it is hard to make a comprehensive statement. Otherwise this is an inspiring paper.

A few things still need some improvement.

I am missing one very simple line of control experiments explaining the side reaction of fuel 1 (DVB) with fuel 2 (NH₂-4 and SH₂-3) in comparison with the substrate tAM 1 and tAM 2. Figure 2. Please provide kinetic data and conversion plots in the SI describing the reaction of DVB with tAM1 and tAM2 in comparison to DVB with tAM1 and tAM2. In parts the latter data are available in Figure 2. I think this data will greatly improve the understanding on describing side reactions. I wonder also about the Michael addition side reaction at the double bond of DVB. What is happening there and which role does it play overall.

The introduction is far too generic. No specific background is given. For instance work on chemical fuels and polymers is basically not described, but the references appear even in the main text (e.g. the key reference 55 and 56). This does not help any unqualified reader in the sense of understanding specific background, and is also partly misleading regarding novelty and state of the art. Why should the reader be informed about the many studies on all kinds of self-assemblies, but the key part on polymers and block copolymers is left out from the introduction? The key part on polymers should be described in greater depth, while less relevant studies can be summarized more coarsely. Nat. Comm. Offers enough space.

I believe it would be sensible to point to the shortcoming of the CRN. It presently takes days to operate a transient cycle (reviewer 3 pointed this out correctly). Can this be improved and how can this be improved? This would add a good discussion part in the conclusion. Overall that means, the CRN is at present not highly applicable as claimed by the authors, simply because it cannot be tuned easily in the lifetime.

Reviewer #3:

Remarks to the Author:

The authors have done an excellent job in answering the reviewer's questions, and addressing their concerns in the revised version of this manuscript.

In particular, the scope, potential and novelty of this work is now presented much clearer in a context of other approaches and concepts that have been published in the literature. Furthermore, the authors clarified the role of the waste product in the cycling of micelle assembly/disassembly.

Also, the authors provided sufficient justification in their rebuttal why some of the experiments suggested by the reviewers were not feasible or out-of-scope.

Together, I can recommend this work now for Nature Communications.

We thank the reviewers for their constructive comments and have amended the work and manuscript accordingly. Including additional experiments, this has led to some changes in the manuscript and supporting information, which are detailed below.

Referee #2

Reviewer #2 (Comments for the Author):

Specific comments:

A few things still need some improvement.

I am missing one very simple line of control experiments explaining the side reaction of fuel 1 (DVB) with fuel 2 (NH₂-4 and SH₂-3) in comparison with the substrate tAM 1 and tAM 2. Figure 2. Please provide kinetic data and conversion plots in the SI describing the reaction of DVB with tAM1 and tAM2 in comparison to DVB with tAM1 and tAM2. In parts the latter data are available in Figure 2. I think this data will greatly improve the understanding on describing side reactions.

In the previous version of the manuscript, we described control experiments regarding the reaction of DVP with S,N-terminal nucleophiles (SH-3 and NH₂-4), which were available in the SI, Figure S2.2 for DVP+SH-3 (ratio = 1:1) and Figure S2.3 for DVP+NH₂-4 (ratio = 1:4) and the conversion plot was presented in Figure S2.4. We now provide further experimental data to the reader by direct comparison of the reaction between DVP and t-Am-1/2 to DVP and SH-3/NH₂-4. As reviewer 2 suggested, we use the data which was available from Figure 2 for the DVP and t-Am-1/2 reaction. The ratio used in the experiment was DVP (1.0 eq.) together with t-AM-1 or 2 (0.2 eq.). We performed the additional control experiments under the same conditions: DVP (1.0 eq.) and SH-3 or NH₂-4 (0.2 eq.) and followed the reactions by ¹H NMR and then plotted all reactions for comparison (see figure below).

Rebuttal figure 1: Conversion plot over time for the reaction of DVP (1.0 eq.) with t-AM-2 (0.2 eq.), t-AM-1 (0.2 eq.), SH-3 (0.2 eq.) and NH₂-4 (0.2 eq.). All measurements were done in duplicate. Solid lines represent the pseudo first order kinetic model fit to the experimental data.

Furthermore, since these reactions were performed under pseudo-first order conditions by using one of the reactants in excess (DVP), we were able to compare these reactions based on their reaction rate constants. The reaction order constants were calculated by fitting the conversion of DVP over time with Eq.S3:

$$\ln\left(\frac{[B]_t}{[B]_0}\right) = -k[A]_0t \quad \text{Eq.S3}$$

This allows us to not only qualitatively compare all reactions (NMR plots) but also quantitatively based on their reactivity (reaction constants) as shown in Rebuttal Table 1.

Rebuttal Table 1.: Summary of the reaction rates for DVP with t-AM-1, t-AM-2, SH-3 or NH₂-4.

A			
B			
System	Reaction	k (M ⁻¹ ·h ⁻¹)	R ²
A	DVP + t-AM-1	43.5 ± 3.21	0.996
	DVP + t-AM-2	5.14 ± 0.62	0.992
B	DVP + SH-3	20.3 ± 4.74	0.982
	DVP + NH ₂ -4	0.27 ± 0.02	0.995

* Conditions: (A) 0.04 mM of DVP, 0.008 mM tertiary amine (t-AM-1 or t-AM-2) in 0.1 M phosphate buffer (pH 7.4), 25 °C, (B) 0.04 mM of DVP, 0.008 mM nucleophile (SH-3 or NH₂-4) in 0.1 M phosphate buffer (pH 7.4), 25 °C.

We added all new data and findings in the SI (Figure S2.1), as suggested by reviewer 2, and direct the reader from the manuscript to the SI data.

from the manuscript:

We conducted a reaction rate study to further understand the reactivity of DVP towards the tertiary amines and nucleophiles with their order being t-Am-1 > SH-3 > t-Am-2 >> NH₂-4 (Figure S2.1). As t-Am-1 ($N = 18.80$ in CH₃CN⁵⁵) is more nucleophilic than t-Am-2 ($N = 12.90$ in CH₂Cl₂⁵⁵), we can attribute these kinetic variations to the difference in nucleophilicity of the employed tertiary amine.⁵⁶ Hence, it was not surprising that complete conversion of DVP to DVP-t-Am-1 is on average $\sim 9.0 \pm 0.8$ fold faster than that of DVP-t-Am-2 (Figure 2b/c). Similarly, for the progression of the deactivation reaction, DVP-S formed on average $\sim 30 \pm 3.4$ times faster using t-Am-1 than when using t-Am-2 (Figure 2b/c). The blank reaction of DVP with thiol (no tertiary amine) using a ratio of 1:1, takes ~ 110 hours to reach completion (SI, Figure S2.2 & S2.4). Addition of thiol to a mixture of activated intermediate in excess DVP predominantly leads to reaction with the activated intermediate (Figure 2b,c). This observation confirms that the reactivity of thiol with the activated intermediate (DVP-t-Am-1 or DVP-t-Am-2) is kinetically highly favoured over the background reaction with DVP. Although reaction kinetics are amine dependent, in both cases the formation of waste product follows quantitatively after each SH-3 addition event (Figure 2b/c), which confirms the absence of unwanted SH-3 side reactivity such as disulfide formation.

I wonder also about the Michael addition side reaction at the double bond of DVB. what is happening there and which role does it play overall.

Based on the NMR plots in Figure S2.2 for DVP+SH-3 (ratio = 1:1) and figure S2.3 for DVP+NH₂-4 (ratio = 1:4) and after careful evaluation of the NMR spectra, no evidence of Michael addition products can be found. As seen in the spectra, the reactions run clean with visible S_N2'-mediated nucleophilic substitution products. Therefore, we conclude that the Michael addition does not play a role during the signal-induced cycle.

Similarly, no Michael products were found using NH₂-4 as nucleophile in excess (4x). Indeed, threonine (NH₂-4) is a very weak nucleophile, making any Michael addition for the autonomous cycle system highly unlikely and hence such side reactions do not play a role in the overall conversion of DVP in this system.

The introduction is far too generic. No specific background is given. For instance work on chemical fuels and polymers is basically not described, but the references appear even in the main text (e.g. the key reference 55 and 56). This does not help any unqualified reader in the sense of understanding specific background, and is also partly misleading regarding novelty and state of the art. Why should the reader be informed about the many studies on all kinds of self-assemblies, but the key part on polymers and block copolymers is left out from the introduction? The key part on polymers should be described in greater depth, while less relevant studies can be summarized more coarsely. Nat. Comm. Offers enough space.

We thank the reviewer for the comment and made appropriate adjustments and included references 55 and 56 into the introduction.

From the amended manuscript 'introduction':

(...) "A wide variety of material activation processes have been described in literature, which are typically powered by various chemical fuels (e.g. carbodiimide fuels [15], methylation agents [5], redox-reagents [16,17] or thioesters [18]) or physical stimuli, such as light or ultrasound to name a few. [19–21]" (...)

(...) So far, studies of transient materials have focused on applying existing CRNs to different material classes, frequently composed of low molecular weight amphiphiles [20,32,35,36], gelator assemblies [10,11,29], nanoparticles [23] or (block)-copolymers [13,16,17,22,24,37–41]. In particular for polymer materials, established (de)activation strategies are generally used (e.g. pH change, carbodiimide/hydrolysis, redox-reagents or light). In contrast, new CRNs are commonly developed for a specific material class or connected to the design of the molecular building blocks. In consideration of current methods for the development of materials with autonomous behaviour and pre-programmed response, we lack a general framework. [42] Thus, the development of scalable CRNs and their applicability to a variety of different materials for the design of interactive structures is an extremely attractive challenge. (...)

The introduction now has additional background on chemical fuels and literature examples on non-equilibrium CRNs applied on polymers, as suggested by the reviewer.

I believe it would be sensible to point to the shortcoming of the CRN. It presently takes days to operate a transient cycle (reviewer 3 pointed this out correctly). Can this be improved and how can this be improved? This would add a good discussion part in the conclusion. Overall that means, the CRN is at present not highly applicable as claimed by the authors, simply because it cannot be tuned easily in the lifetime.

The authors thank reviewer 2 for the comment and suggestion on improving the paper. Indeed, the transient cycle takes days to complete. This deficiency is partially related to the rather weak EWG = $(\text{EtO})_2\text{P}=\text{O}$ on the allyl substrate, which dictates the reaction speed through the $\text{S}_{\text{N}}2'$ mechanism and ultimately its lifetime. To improve the lifetime of the CRN, substrates with stronger EWG could be used (e.g., EWG = ester).

After having developed the main concepts of the current manuscript, our group continued to work on CRNs based on allylic substitution reactions. In a subsequent paper (DOI: 10.1039/D2SC00805J = reference 67), we present a CRN based on a more reactive substrate (EWG = methylester) and apply the CRN to complex coacervate micelles. The paper shows that by varying the substrate (from EWG = diethylphosphonate to methylester), essentially the CRN lifetime can be tuned and the overall time of the CRN run can be greatly reduced. Overall this means, that for the CRN in the current manuscript the lifetime of the cycle can be tuned by varying the substrate (using stronger EWG allyl acetates).

We have amended the manuscript 'conclusion' accordingly:

(...)” The principle of temporally programming the behaviour of synthetic materials, shown in this work, is applicable not only to constructing out-of-equilibrium synthetic structures but also as advanced strategy for controlled molecular cargo delivery. By using the underlying CRN, its scalability and applicability to different material classes combined with its possibilities of tuning the kinetic profile based on choice of nucleophile or choice of fuel [67], a variety of new designs for interactive structures can be envisioned. We therefore anticipate that this concept will contribute to the development of next generation soft materials, where signal or time-programmed control over charge density allows for interactive and adaptive material properties, such as stiffening, adhesion or motion.” (...)

Reviewers' Comments:

Reviewer #2:

Remarks to the Author:

COngratulations to the authors for going the extra mile and making these last improvements. all perfect now. great paper.